# Face masks reduce emotion-recognition accuracy and perceived closeness

**Felix Grundmann⊙\*, Kai Epstude, Susanne Scheibe⊙**

Department of Psychology, University of Groningen, Groningen, The Netherlands

\* f.u.grundmann@rug.nl

## Abstract

Face masks became the symbol of the global fight against the coronavirus. While face masks' medical benefits are clear, little is known about their psychological consequences. Drawing on theories of the social functions of emotions and rapid trait impressions, we tested hypotheses on face masks' effects on emotion-recognition accuracy and social judgments (perceived trustworthiness, likability, and closeness). Our preregistered study with 191 German adults revealed that face masks diminish people's ability to accurately categorize an emotion expression and make target persons appear less close. Exploratory analyses further revealed that face masks buffered the negative effect of negative (vs. non-negative) emotion expressions on perceptions of trustworthiness, likability, and closeness. Associating face masks with the coronavirus' dangers predicted higher perceptions of closeness for masked but not for unmasked faces. By highlighting face masks' effects on social functioning, our findings inform policymaking and point at contexts where alternatives to face masks are needed.

## Introduction

COVID-19 has created a new normal, changing how people interact in fundamental ways. While people still frequently engage with unknown others, be it in the grocery store or on the bus, many interactions are now between strangers wearing masks. With more than 24 million confirmed cases and more than 250 thousand daily new cases worldwide [1], 50+ countries require their citizens to wear face masks in public [2]. While face masks effectively reduce the risk of infection [3], they may impact encounters between strangers in meaningful ways. Frequently voiced concerns pertain to difficulties in reading emotions in masked faces and to related disturbances in social interactions [4]. In the present experimental study, we sought to evaluate whether such concerns are justified. Drawing from socio-functional theories of emotion [5,6] and research on the rapid formation of trait impressions based on facial cues [e.g., a smile or youthful features; 7,8], we examined face masks' effect on emotional and social inferences. Specifically, we investigated whether the reduction of facial cues due to wearing a face mask undermines emotion-recognition accuracy and perceptions of trustworthiness, likability, and closeness. We further explored how the valence of the emotional expression and mask-related associations interact with face masks to influence these social judgments. Showing that

**Data Availability Statement:** All data are accessible online via the OSF (https://osf.io/av9gp/?view_only=c42719b123b54d76b0fbcae0c073fdd6).

**Funding:** The authors received no specific funding for this work.

**Competing interests:** The authors have declared that no competing interests exist.

face masks influence social functioning is of great practical interest given the prevalence of face masks in everyday life. Moreover, as accurate emotion recognition and positive social inferences are crucial in contexts where establishing relationships is pivotal, our study may highlight the need for alternatives to face masks and thereby inform policy making and political debate.

## Face masks and emotion recognition

Effective human communication relies on the accurate perception of emotions [9]. For instance, which emotions people perceive influences how they respond to their interaction partners [10]. According to the emotion-as-social-information model [6], an emotional expression can instigate inferential processes, which inform cognitions and actions in turn. For example, if a target person expresses sadness when telling an observer about not being able to reach the top of the shelf, observers may respond by offering their help. Similarly, observers may decide to give up their seat for an older person when they notice other passengers are angrily looking at them. As different emotions convey different information, the accurate recognition of emotions is crucial. If emotion recognition fails, interaction partners may act in inappropriate ways, ultimately undermining the success of the interaction.

Accurate emotion recognition also plays a pivotal role in social interactions for another reason. Keeping with the emotion-as-social-information model [6], recognizing a target person's emotions affords observers with the possibility to mimic their affective expression [see also 11]. Emotional mimicry–matching the nonverbal behaviors underlying emotional expressions [12,13]–serves an affiliative function and increases liking, facilitating the establishing of a relationship in turn [13,14]. Hence, being able to accurately categorize the emotional expression of an interaction partner is particularly important when people meet for the first time.

People express what they feel using various modalities including their face [15]. The face constitutes an especially valuable source of affective information, as emotions recruit facial muscles in unique ways [16–18]. Hence, to infer the emotional experiences of others, observers need to decode information conveyed by facial cues. Information-rich areas are the eyes, nose, and mouth. Importantly, prototypical face masks reduce the number of available facial cues by covering the mouth and part of the nose (when worn correctly). Thereby, they may undermine observers' ability to correctly recognize emotions expressed in a target person's face [19,20]. Indeed, adult participants exhibited lower accuracy rates for faces whose mouth regions were covered [vs. not; 20].

Prior research highlighted demographic factors which (may) influence emotion-recognition accuracy in addition to face masks. These influences are thought to arise in part from sex-related differences in brain activity as well as aging-related changes in the brain and physiognomy. For instance, different brain areas are activated in men and women when processing emotional material [e.g., 21,22]. Such differences in brain activation are thought to affect emotion recognition. Indeed, there is meta-analytic evidence of women having superior emotion perception relative to men [23,24, but see also 25].

Focusing on age-related changes in the brain, Ruffman and colleagues [26] argued that differences in neuron density and neurotransmitter balance best account for lower emotion-recognition accuracy among older (vs. younger) adults. For instance, when asked to classify emotional expressions depicted in portrait photographs, older adults consistently performed worse than younger adults [e.g., 26]. Importantly, age differences in emotion recognition may be even more pronounced when target persons wear face masks because the lack of facial cues makes the task more difficult. Research suggests that discrepancies in performance on cognitively taxing tasks between younger and older adults widen when task difficulty increases [27].

There is also evidence that increases in task difficulty negatively impact emotion perception for older adults [28]. Hence, neurological aging-related differences may be particularly consequential when demands are high [26].

Another age-related factor impacting emotion-recognition accuracy is the target's age. Reductions in the tautness of the skin with increased age render emotional expressions more ambiguous, affecting emotion perception in turn. Compared to young adult faces, folds and wrinkles increase the difficulty of decoding expressed emotions on old adult faces [29]. When wearing face masks, the additional uncertainty may amplify emotion-recognition difficulties for old (vs. young) target faces. As both, age-related changes in faces' morphology and face masks, reduce facial cues, the number of available facial cues may be too low for accurate emotional decoding.

Together, we tested the following hypotheses:

H1: Wearing a face mask (vs. not) negatively impacts emotion-recognition accuracy.

H2: Women are more likely than men to correctly recognize emotional expressions.

H3: Young adults are better at recognizing emotional expressions than old adults.

H4: Emotion-recognition accuracy is lower for old relative to young target faces.

H5: The difference in the likelihood of accurate emotion recognition for young and old observers is larger when the target face is masked (vs. unmasked).

H6: The difference in the likelihood of accurate emotion recognition for young and old target faces is larger when the target face is masked (vs. unmasked).

## Face masks and social judgments

When interacting with strangers, people have little to no information about what to expect from the other. To increase the predictability of their actions [30], people use facial cues to make trait judgments [8,31,32]. For example, people deduct a stranger's trustworthiness and likability from their face [7]. Specifically, the perceived health, familiarity, emotional expression, and childlikeness of a face meaningfully shape rapid trait inferences [8]. However, when an unknown interaction partner is wearing a face mask, important information about the person's character become unavailable. The associated increase in uncertainty may in turn impact social judgments [33]. Indeed, different studies have linked familiarity, ambiguity, or uncertainty to lower perceptions of trustworthiness, likability, and closeness [e.g., 34–36].

Regarding perceived trustworthiness, Acay-Burkay and colleagues [34] showed that unfamiliarity (vs. familiarity) negatively impacted trust in a negotiation partner (Study 2). Similarly, participants high (vs. low) in intolerance of ambiguity trusted interaction partners less in a trust game [37]. Moreover, the amount of information available about an unknown other predicted the extent to which they were perceived as trustworthy [38].

Regarding perceived likability, when two strangers meet for the first time, the more familiar they become the more they like each other [35]. Hence, it seems that increases in information about another person promotes liking [but see also 39]. Indeed, uncertainty negatively correlated with social attraction following offline interactions [40]. Similarly, for online interactions, reductions in uncertainty predicted increased social attraction [41].

Regarding perceived closeness [akin to psychological distance; 42], face masks may not only separate observers and targets in a figurative sense but also reduce perceptions of closeness via uncertainty. Indeed, reductions in uncertainty predicted increased intimacy [36]. Moreover, decoding accuracy is higher for dyads whose relationship is relatively more intimate [e.g., 43].

Together, we tested the following hypotheses:

H7a: People perceive masked (vs. unmasked) targets to be less trustworthy.

H7b: People perceive masked (vs. unmasked) targets to be less likable.

H7c: People perceive masked (vs. unmasked) targets to be less close to them.

**Exploring emotional valence and context.**   Beyond main effects of face masks on social judgements, we further explored how the valence of the target's emotion in combination with (vs. without) a face mask influences the observer's perceptions of trustworthiness, likability, and closeness. According to the emotion-as-social-information model [6], observers use the valence of facial emotional expressions to make trait judgments [e.g., 44,45]. For example, relative to happy and surprised faces, observer deemed angry faces more hostile and less trustworthy [46]. These inferences are partly based on what emotions signify. In general, positive emotions such as happiness signal an approach orientation focused on social engagement while negative emotions such as anger signal an avoid orientation focused social disengagement [47, but see also 48]. In other words, the valence of a target's emotional expression may have downstream consequences for observers' social judgments [6,8].

We also explored contextual effects. Face masks are meaning-laden and may thus shape inferential processes [e.g., 49]. On the one hand, face masks entered public life on a global scale (and beyond religious applications) only during the COVID-19 pandemic, serving as a reminder of the virus' threat. On the other hand, mandating face masks in public has meant the partial return to pre-pandemic life. Thus, face masks may also symbolize increased personal freedom and opportunity. Based on these considerations, we examined mask-as-threat and mask-as-opportunity associations together with the expressed emotion's valence and face masks in the context of the social judgments.

## Method

### Participants

We derived our target sample size by reference to similar prior studies. For example, Kret and Fischer [50] recruited 134 participants. To safeguard against loss of power due to preregistered participant exclusions, we aimed for a total sample size of 200.

We commissioned the panel survey company Respondi with the data collection. They recruited 201 participants living in Germany. As preregistered, we excluded participants who failed the attention check ($n = 9$) or indicated dishonest responding ($n = 1$), yielding a final sample size of 191 (52.9% female). The majority was born in Germany (93.7%) and reported German as their only ethnicity (90.1%).

We utilized stimuli from the FACES database [51; further described below]. Stimuli are portrait photographs of German adults belonging to three age segments (young: 19–31 years; middle-aged: 39–55 years; old: 69–80 years). Participants were recruited to belong to these age groups; 60 were young adults ($M = 24.00$, $SD = 3.14$, range = 19.00–31.00), 65 were middle-aged ($M = 48.80$, $SD = 4.58$, range = 40.00–55.00), and 66 were older adults ($M = 72.50$, $SD = 2.34$, range = 69.00–79.00). Within age segments, we approximated an equal female-male gender split (between 46.2% and 59.1% were female).

### Materials and procedure

The study was approved by the Ethical Committee of Psychology of the authors' university (research code: PSY-1920-S-0480). It was conducted online from June 25 until July 8 and

programmed in Qualtrics (https://www.qualtrics.com). Participation was rewarded with 1€. Participants were (without their awareness) randomly assigned to one of two conditions: In the *control* condition, participants saw original face stimuli; in the *mask* condition, target faces were covered by a prototypical face mask. After giving active informed consent and agreeing to not download the study materials, participants provided basic demographic information. Subsequently, participants completed an emotion-recognition task and provided social judgments as described below. They then completed measures of their mask-as-threat and mask-as-opportunity associations. We also assessed participants' preoccupation with COVID-19 and their exposure to face masks. To this end, we developed four face-valid items to measure preoccupation with COVID-19 and adapted items by Wang and colleagues [52] to measure exposure to face masks. As these scales were not relevant for the analyses, we do not discuss them further. However, detailed information about the scales including the items can be found in the online supplemental material (https://osf.io/av9gp/?view_only=c42719b123b54d76b0fbcae0c073fdd6).

The emotion-recognition task consisted of 38 trials (see Fig 1 for the trial structure). Following two practice trials to ensure task comprehension, participants completed 36 critical trials. In each trial, participants were first shown a portrait photograph of an adult face for two seconds. We obtained permission to retrieve and adapt these photographs from the FACES database [51]. As all photographs in the databank depict native German community members, FACES stimuli were particularly suited for the purpose of the present investigation [52]. The stimuli used in the critical trials systematically differed in age (young, middle-aged, old), gender (female, male), and expressed emotion (neutral, happy, fearful, angry, sad, disgusted). Factors were fully crossed. We randomized the order of the critical trials. FACES stimulus codes are included in the online supplemental material.

After the stimulus disappeared, participants selected the expressed emotion from an emotion list. In addition to all displayed emotions (neutral, happy, fearful, angry, sad, disgusted), the emotion list contained three distractor options (surprised, proud, amused) to prevent ceiling effects. We coded correct choices as 1 and incorrect choices as 0 ($n_{missing}$ = 2).

Next, participants judged the extent to which they deemed the target trustworthy, likable, and close to them on a 5-point Likert scale (ranging from '1 = absolutely not' to '5 = absolutely'). High scores reflected greater perceived trustworthiness ($n_{missing}$ = 2), likability ($n_{missing}$ = 1), and closeness.

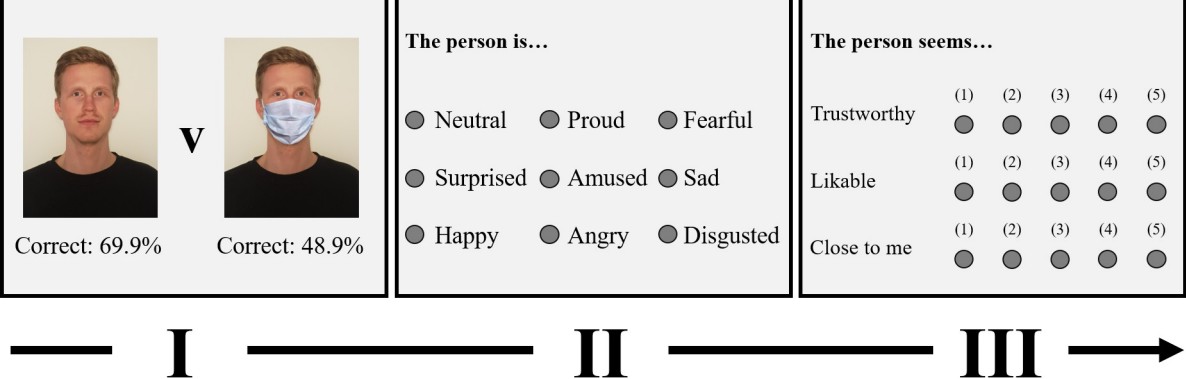

**Fig 1. Trial structure of the emotion-recognition and social judgment task.** Each square represents a single page on participants' screen. Please note that the stimuli and the item wording are not accurate but serve illustrative purposes. For copyright reasons, we cannot include the stimuli used in the study. Percentages below the included stimuli indicate the number of correctly recognized emotional expressions across participants and trials in each condition. The depicted individual has given written informed consent (as outlined in the PLOS consent form) to be included in the figure.

After the emotion-recognition task, we assessed participants' mask-as-threat and mask-as-opportunity associations using three statements each. The statements were developed for current purposes. For all randomly ordered statements, participants indicated their agreement on a 5-point Likert scale (ranging from '1 = completely disagree' to '5 = completely agree'). The items used to measure mask-as-threat associations were 'I think about the dangers of Corona when I see others wearing protective face masks', 'Protective face masks remind me that the threat posed by Corona is still real', and 'When I see others wearing protective face masks, I feel safe' (reverse-coded). The items used to measure mask-as-opportunity associations were 'Protective face masks allow me to interact with others', 'Society can start returning to normal thanks to wearing protective face masks', and 'What I can do is limited despite the protective face masks' (reverse-coded). Given the problematic reliability estimates (threat: α = -0.24; opportunity: α = 0.65), we dropped the reverse-coded items. This resulted in satisfactory reliability (threat: α = 0.76; opportunity: α = 0.77). We averaged participants' scores for each association type. Higher scores reflect stronger mask-as-threat and mask-as-opportunity associations, respectively.

## Analytic strategy

We preregistered all hypotheses and confirmatory statistical tests prior to data collection (https://aspredicted.org/blind.php?x=2td8ze). Given the hierarchical structure of the data (trials nested within persons), we relied on multilevel (logistic) regression models. We used R for all analyses [53]. For model fitting, we utilized the glmer and lme functions from the lme4 [54] and nlme [55] packages. For the multilevel logistic regression analyses, we used the optimx optimizer (method = nlminb) from the optimx package [56] to ensure model convergence. Prior to all confirmatory analyses, we fitted empty models to compute the intra-class correlation [ICC; 57]. The ICC captures the amount of variance in the outcome accounted for by the level-2 units. To compare model fit, we relied on deviance tests. Predicted probabilities are based on models' average intercept. All data are accessible online (https://osf.io/av9gp/?view_only=c42719b123b54d76b0fbcae0c073fdd6). Unlike preregistered, we did not exclude any observations (see the online supplemental material for details).

To investigate the effect of target- and observer-dependent factors and their interactions on emotion-recognition accuracy (H1-H6), we estimated two multilevel logistic regression models. The first model included emotion-recognition accuracy as the dependent variable at level 1. Condition, gender, age (all at level 2), and age-of-target (level 1) served as predictors. Participants' age and age-of-target were each represented by two dummy variables coded as 010 and 001 for young, middle-aged, and old. Unlike preregistered (see the online supplemental material for details), only the intercept randomly varied at level 2. The second model included interactions between condition and age and between condition and age-of-target. Hypotheses were tested by comparing the models' fit and by examining the coefficients of Model 2. Note that one may argue that expressions of happiness and amusement resemble each other to such an extent that classifying a happy target as amused reflects an accurate appraisal. Analyses in which we treat such classifications as correct can be found in the online supplemental material.

To investigate condition effects on perceived trustworthiness, likability, and closeness (H7a-H7c), we fitted two multilevel regression models per outcome (level 1). The first model constituted an empty model which we used to estimate the ICC. We extended this model by adding condition as a predictor at level 2. Hypotheses were tested by comparing the models' fit and by examining condition effect's direction in Model 2.

As preregistered, we conducted exploratory analyses. We examined the individual and joint effects of condition, the emotion's valence, as well as mask-as-threat and mask-as-opportunity associations on the social judgments. For valence, we coded happy and neutral expressions as 0, indicating non-negative expressions, and disgusted, sad, fearful, and angry faces as 1, indicating negative emotional expressions. Continuous variables were grand-mean centered to ease interpretation. For each outcome, we fitted two multilevel regression models. The intercept varied randomly at level 2 in all models. The first model included condition (level 2), valence (level 1), and their interaction as predictors. The second model extended Model 1 by adding mask-as-threat and mask-as-opportunity associations (level 2) and their interactions with condition as predictors. If Model 2 fitted the data significantly better than Model 1, we retained it; otherwise, we reported Model 1.

## Results

Overall emotion-recognition accuracy declined from 69.9% for unmasked to 48.9% for masked target faces (see also Fig 2). Women (62.8%) tended to outperform men (55.2%). Regarding age-of-target, accuracy was highest for young faces (61.4%) and lowest for old faces (56.9%), with middle-aged faces falling in-between (59.4%). Similarly, young and old participants achieved the highest (64.6%) and lowest (52.2%) average accuracy, respectively, with middle-aged participants falling in-between (59.4%). Consistent with that, age negatively correlated with participants' mean accuracy across trials ($r = -.27$, $p < .001$; see Table 1 for all correlations and descriptive statistics). Mask-as-threat and mask-as-opportunity associations strongly covaried ($r = .43$, $p < .001$). Mask-as-opportunity associations were significantly stronger in the control ($M = 3.59$, $SD = 0.96$) versus mask ($M = 3.21$, $SD = 1.12$) condition, $t = 2.49$, $df = 186.10$, $p = .014$, 95% CI [0.08, 0.67], $d = .36$ (two-sided test). There was no evidence for differences in mask-as-threat associations between the control ($M = 3.55$, $SD = 1.00$)

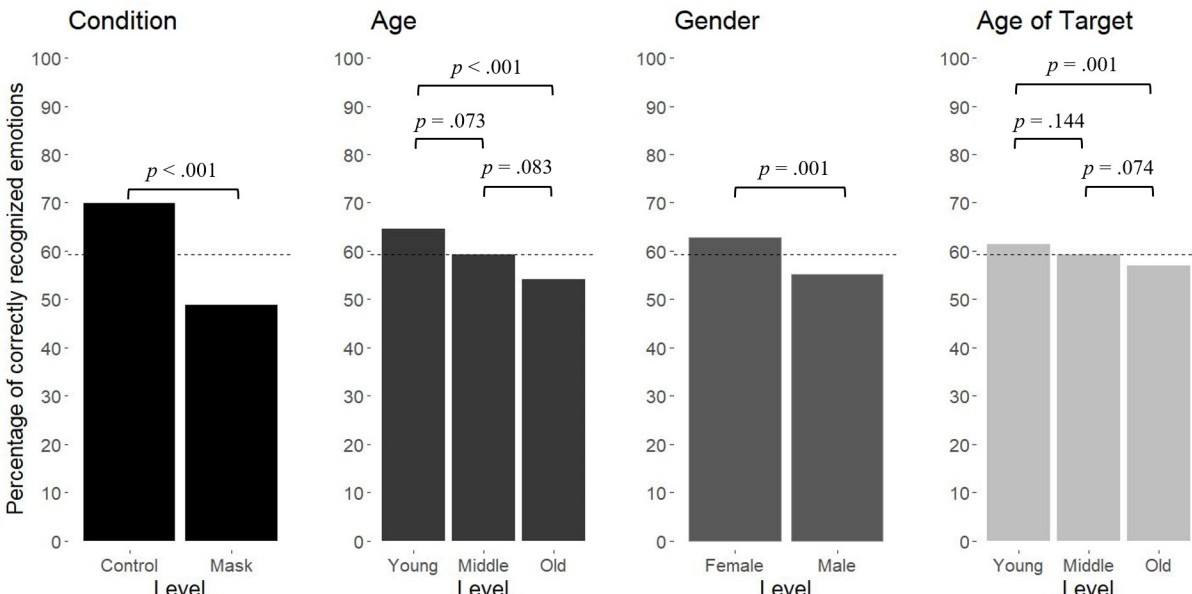

**Fig 2. Factor-based comparisons of emotion-recognition accuracy.** Comparisons are based on simple multilevel logistic regression models with emotion-recognition accuracy as the outcome and the respective factor as the sole predictor. For factors with three levels, we repeated the analysis using the level initially coded as 010 as the reference group. The dashed line indicates the average emotion-recognition accuracy across trials, participants, and factors ($\hat{\rho}_| = .12$).

**Table 1. Associations between all continuous variables included in the study.**

| Variable | M (SD) | Range | 1 | 2 | 3 | 4 | 5 | 6 | 7 |
|---|---|---|---|---|---|---|---|---|---|
| 1. Age | 42.20 (20.00) | 19.00–79.00 | - | .002 | .068 | < .001 | .029 | .095 | .025 |
| 2. Mask-as-threat | 3.55 (1.00) | 1.00–5.00 | .22* | - | < .001 | .313 | .064 | .489 | .085 |
| 3. Mask-as-opportunity | 3.40 (1.05) | 1.00–5.00 | .13 | .43** | - | .838 | .109 | .243 | .016 |
| 4. Average accuracy (across trials) | 0.59 (0.17) | 0.14–0.91 | -.27** | -.07 | -.01 | - | .777 | .888 | .196 |
| 5. Average perceived trustworthiness (across trials) | 3.03 (0.51) | 1.00–5.00 | .16* | .13 | .12 | .02 | - | < .001 | < .001 |
| 6. Average perceived likability (across trials) | 2.87 (0.53) | 1.11–4.14 | .12 | .05 | .08 | .01 | .79** | - | < .001 |
| 7. Average perceived closeness (across trials) | 2.58 (0.65) | 1.00–4.06 | .16* | .12 | .17* | -.09 | .55** | .79** | - |

*Note.* Pearson product-moment correlations are displayed in the bottom triangle, with the associated *p*-values displayed in the top triangle.

* *p* < .05

** *p* < .001.

and mask (*M* = 3.56, *SD* = 1.01) conditions, *t* = -0.10, *df* = 188.97, *p* = .923, 95% CI [-0.30, 0.27], *d* = .01 (two-sided test).

## Testing effect of condition, age, gender, and age of target on accuracy

We fitted two multilevel logistic regression models to test Hypotheses 1–6 (see Table 2). Compared to the main-effects model including condition, age, age-of-target, and gender as predictors, adding interactions between condition and age as well as between condition and age-of-target marginally improved the fit of the model, $\chi^2(4) = 8.42$, *p* = .074. About 12% of the variance in emotion-recognition accuracy was accounted for by the level-2 units. The predicted probability of accurately recognizing the emotion as a function of the predictors is illustrated in Fig 3.

Consistent with H1, the mask condition led to a significant decrease in emotion-recognition accuracy relative to the control condition. The probability of accurate recognition for young females seeing a young face was expected to drop from 77.8% for unmasked faces (reference category) to 61.5% for masked faces. In line with H2, being old (vs. young) predicted a significant decrease in emotion-recognition accuracy. Compared to the reference category, the probability of accurately recognizing emotions was expected to decline to 72.1% for old females. Supporting H3, being male (vs. female) predicted a significant decline in emotion-recognition accuracy. The probability of accurately recognizing emotions was expected to decrease to 73.4% for young males relative to the reference category. As anticipated (H4), emotion-recognition accuracy declined for old (vs. young) target faces. Compared to the reference category, the probability of accurate recognition was expected to drop to 72.3% for unmasked old faces.

The predicted interaction effect between the face being masked (vs. not) and old (vs. young) age (H5) just failed to reach statistical significance, *b* = -0.38, *p* = .057. Nevertheless, the difference in the expected probability of correct recognition with and without taking the interaction into account equaled 9.5%, which is considerable. Hypothesis 6 was not supported (see Table 2). We found no evidence that the negative effect of seeing an old (vs. young) target face on emotion-recognition accuracy was further exacerbated by face masks (vs. no face masks).

## Testing the effect of condition on social judgments

We fitted two multilevel regression models for perceived trustworthiness, likability, and closeness to test H7a-H7c. For each outcome, the ICC was considerably larger than zero (trustworthiness: $\hat{\rho}_| = 0.26$; likability: $\hat{\rho}_| = 0.24$; closeness: $\hat{\rho}_| = 0.34$).

**Table 2. Multilevel logistic regression models capturing the effect of condition, age, gender, and age of target on emotion-recognition accuracy.**

| | Model 1 | | | | Model 2 | | | |
|---|---|---|---|---|---|---|---|---|
| Fixed effect | B (SE) | z (p) | 95% CI | OR | B (SE) | z (p) | 95% CI | OR |
| Intercept | 1.32 (0.10) | 13.31 (< .001)** | 1.11, 1.51 | 3.73 | 1.25 (0.12) | 10.10 (< .001)** | 1.00, 1.50 | 3.50 |
| Condition | | | | | | | | |
| Mask | -0.91 (0.08) | -11.01 (< .001)** | -1.09, -0.74 | 0.40 | -0.78 (0.16) | -4.79 (< .001)** | -1.13, -0.44 | 0.46 |
| Age | | | | | | | | |
| Middle-aged | -0.19 (0.10) | -1.85 (.064) | -0.39, 0.02 | 0.83 | -0.16 (0.15) | -1.12 (.263) | -0.44, 0.13 | 0.85 |
| Old | -0.49 (0.10) | -4.90 (< .001)** | -0.71, -0.27 | 0.61 | -0.30 (0.14) | -2.12 (.034)* | -0.59, -0.03 | 0.74 |
| Gender | | | | | | | | |
| Male | -0.23 (0.08) | -2.81 (.005)* | -0.39, -0.06 | 0.79 | -0.24 (0.08) | -2.88 (.004)* | -0.38, -0.072 | 0.79 |
| Age of target | | | | | | | | |
| Middle-aged | -0.09 (0.06) | -1.46 (.143) | -0.20, 0.02 | 0.91 | -0.04 (0.10) | -0.43 (.666) | -0.23, 0.15 | 0.96 |
| Old | -0.21 (0.06) | -3.25 (.001)* | -0.33, -0.08 | 0.81 | -0.29 (0.09) | -3.11 (.002)* | -0.48, -0.09 | 0.75 |
| Condition × Age | | | | | | | | |
| Mask × middle-aged | | | | | -0.05 (0.20) | -0.24 (.811) | -0.47, 0.33 | 0.95 |
| Mask × old | | | | | -0.38 (0.20) | -1.91 (.057) | -0.79, 0.00 | 0.68 |
| Condition × Age of target | | | | | | | | |
| Mask × middle-aged | | | | | -0.09 (0.13) | -0.73 (.463) | -0.33, 0.14 | 0.91 |
| Mask × old | | | | | 0.16 (0.13) | 1.26 (.209) | -0.12, 0.40 | 1.17 |
| Random effect | Parameter | | | | Parameter | | | |
| *Level-two random part*: | | | | | | | | |
| Intercept variance | 0.19 | | | | 0.18 | | | |

*Note*. All available information was used, resulting in a total combined sample size of 6874 trials. The intercept represents the predicted average likelihood of accurate emotion recognition for young female participants seeing young unmasked faces. 95% confidence intervals are based on 500 bootstrap samples.

* $p < .05$

** $p < .001$.

Hypotheses 7a-7c were not supported (see the online supplemental material for all models). Adding condition as a predictor did not improve the model's fit relative to an empty model (trustworthiness: $\chi^2[1] = 0.01$, $p = .952$; likability: $\chi^2[1] = 0.26$, $p = .607$; closeness: $\chi^2[1] = 1.95$, $p = .162$). Thus, the social outcomes were unaffected by face masks (trustworthiness: $b = -0.00$; likability: $b = -0.04$; closeness: $b = -0.13$; but see supplementary analyses accounting for valence below). For all outcomes, intercept variances (trustworthiness: $\tau_0^2 = 0.24$; likability: $\tau_0^2 = 0.25$; closeness: $\tau_0^2 = 0.39$) were smaller than residual variances (trustworthiness: $\sigma^2 = 0.67$; likability: $\sigma^2 = 0.81$; closeness: $\sigma^2 = 0.77$).

## Exploring the effect of condition, valence, and associations on social judgments

We fitted two multilevel regression models for each social judgment (perceived trustworthiness, likability, and closeness). For perceived trustworthiness and likability, Model 2 did not fit the data significantly better than Model 1 (trustworthiness: $\chi^2[4] = 7.41$, $p = .116$; likability: $\chi^2[4] = 5.82$, $p = .213$). For perceived closeness, however, Model 2 fitted the data significantly better than Model 1, $\chi^2(4) = 11.46$, $p = .022$. The final models can be found in Table 3 (see the online supplemental material for all models).

Regarding perceived trustworthiness, participants were predicted to trust unmasked targets less when they expressed a negative (vs. non-negative) emotion ($b = -0.64$, $p < .001$). This

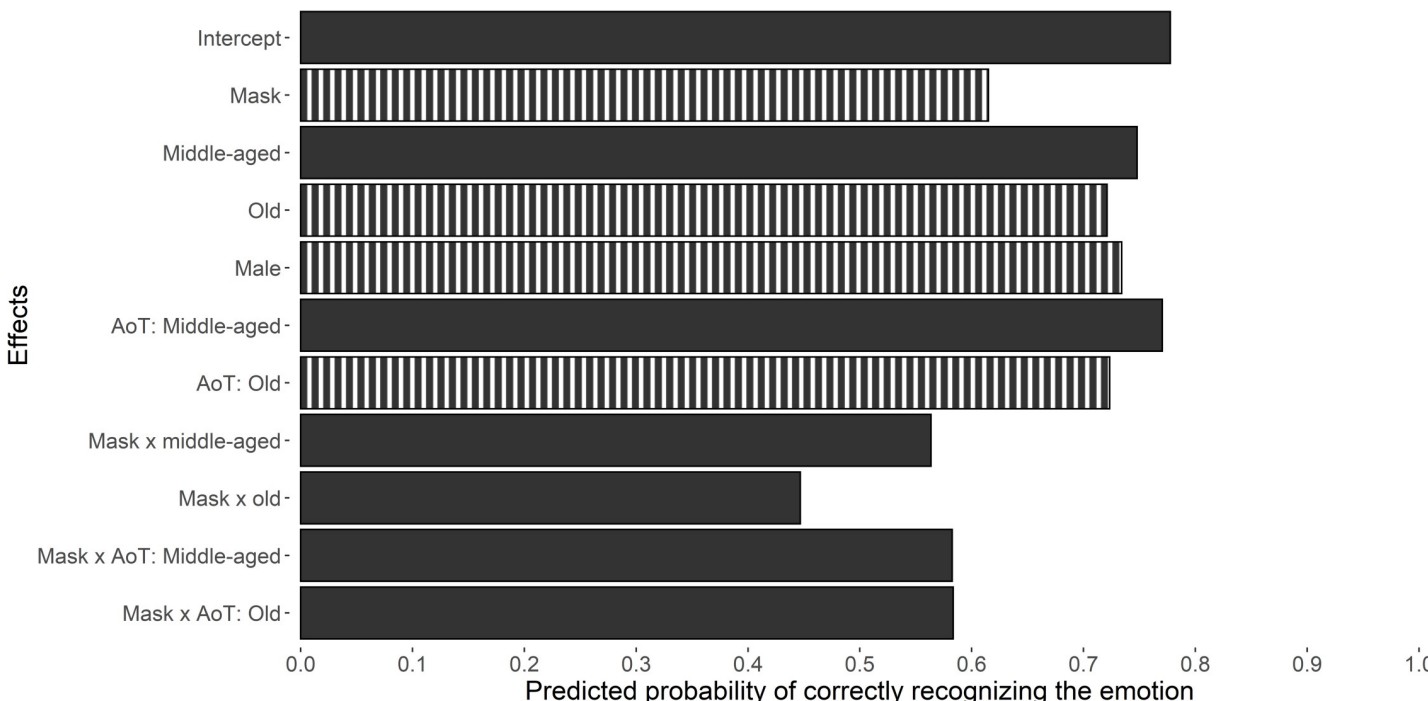

**Fig 3. The effect of condition, age, gender, and age of target on emotion-recognition accuracy.** The predicted effects are shown on the vertical axis, with the expected probability of accurate emotion recognition shown on the horizontal axis. The intercept represents the average expected probability of accurate emotion recognition for young females seeing young and unmasked faces. The ICC equaled .12. The striped pattern indicates a significant effect ($p < .05$). AoT = age of target.

**Table 3. Multilevel regression models capturing the effect of condition, valence, and mask-related associations on social judgments.**

| | Trustworthiness | | | Likability | | | Closeness | | |
|---|---|---|---|---|---|---|---|---|---|
| **Fixed effect** | *B (SE)* | *t (p)* | 95% CI | *B (SE)* | *t (p)* | 95% CI | *B (SE)* | *t (p)* | 95% CI |
| Intercept | 3.46 (0.05) | 62.48 (< .001)** | 3.35, 3.57 | 3.38 (0.06) | 58.65 (< .001)** | 3.27, 3.50 | 3.13 (0.06) | 45.30 (< .001)** | 2.99, 3.27 |
| Mask | -0.09 (0.08) | -1.22 (.223) | -0.25, 0.06 | -0.12 (0.08) | -1.47 (.142) | -0.28, 0.04 | -0.25 (0.10) | -2.56 (.011)* | -0.44, -0.06 |
| Negative valence | -0.64 (0.03) | -22.99 (< .001)** | -0.70, -0.59 | -0.74 (0.03) | -24.27 (< .001)** | -0.80, -0.68 | -0.74 (0.03) | -25.68 (< .001)** | -0.80, -0.68 |
| Mask-as-threat | | | | | | | -0.06 (0.07) | -0.87 (.383) | -0.21, 0.22 |
| Mask-as-opportunity | | | | | | | 0.07 (0.08) | 0.94 (.345) | -0.08, 0.22 |
| Mask × Negative valence | 0.14 (0.04) | 3.46 (.001)* | 0.06, 0.21 | 0.12 (0.04) | 2.80 (.005)* | 0.04, 0.20 | 0.22 (0.04) | 5.17 (< .001)** | 0.14, 0.30 |
| Mask × Mask-as-threat | | | | | | | 0.21 (0.10) | 2.11 (.036)* | 0.01, 0.41 |
| Mask × Mask-as-opportunity | | | | | | | 0.01 (0.10) | 0.12 (.902) | -0.18, 0.21 |
| Random effect | Parameter | | | Parameter | | | Parameter | | |
| *Level-two random part*: | | | | | | | | | |
| Intercept variance | 0.24 | | | 0.25 | | | 0.37 | | |
| *Level-one random part*: | | | | | | | | | |
| Residual variance | 0.59 | | | 0.70 | | | 0.68 | | |

*Note.* The total combined sample size varied between the models due to missing values. All available information was used, resulting in a total combined sample size of 6876, 6875, and 6874 trials for perceived trustworthiness, likability, and closeness, respectively. For perceived trustworthiness and likability as the outcome, the intercept represents the average expected score of participants seeing unmasked faces expressing non-negative emotions. For perceived closeness as the outcome, the intercept represents the average expected score of participants with average mask-related association seeing unmasked faces expressing non-negative emotions.

* $p < .05$

** $p < .001$.

valence effect on trustworthiness was lessened when the target person wore a mask (vs. not; $b = 0.14$, $p = .001$). That is, for target faces expressing a negative emotion, participants were predicted to trust masked targets more than unmasked targets. No main effect of condition was observed, $b = -0.09$, $p = .223$.

The same pattern emerged for perceived likability. Participants were predicted to like unmasked targets less when the expressed emotion was negative (vs. not; $b = -0.74$, $p < .001$). Wearing a mask (vs. not) lessened the effect of valence on likability, $b = 0.12$, $p = .005$. Participants were predicted to like masked targets expressing negative emotions more than unmasked targets expressing negative emotions. No main effect of condition was observed, $b = -0.12$, $p = .142$.

Results were slightly more complex for perceived closeness, as the final model included mask-related associations. Here, the reference category captures the average expected score of participants with average mask-related associations seeing unmasked faces expressing non-negative emotions. Compared to the reference category, participants were predicted to perceive masked targets to be less close, $b = -0.25$, $p = .011$. Similarly, expressing a negative emotion predicted less perceived closeness relative to the reference category, $b = -0.74$, $p < .001$. Moreover, the target being masked reduced the negative effect of negative emotional expressions on perceived closeness, $b = 0.22$, $p < .001$. Finally, the interaction between condition and mask-as-threat associations was positive and significant, $b = 0.21$, $p = .036$. For participants exposed to non-negative expressions and with average mask-as-opportunity associations, stronger mask-as-threat associations predicted higher perceived closeness but only for masked targets.

## Discussion

Many countries mandate face masks in public settings, leading to many interactions between masked strangers. To better understand their collateral consequences for social functioning, we investigated face masks' effects on emotional and social inferences.

### Face masks and emotion recognition

Emotion-recognition accuracy declined for masked (vs. unmasked) faces. This finding is in line with prior studies showing that limiting facial cues reduces the likelihood of accurate emotion recognition [e.g., 20,58 (Study 3)]. Compared with these studies [20], however, overall accuracy for unmasked (69.9% vs. 89.5%) as well as masked (48.9% vs. ~71.7%) targets was considerably lower. The difference might be due to the inclusion of distractor options. Because happiness constituted the only positive target emotion, not including distractor options may have led to biased accuracy estimations. Given that positive emotions other than happiness are encoded in the face [e.g., pride; 59,60], our results may be a better approximation of emotion-recognition accuracy for masked and unmasked faces and stress the need to include distractor options when studying facial expressions of emotion. Nonetheless, selecting a distractor item was never correct in this study, allowing for the possibility that participants learned to classify all positively-valenced target expression as happiness. Hence, the observed overall emotion-recognition accuracy may still be an overestimation.

Consistent with prior findings [e.g., 26,29], we linked lower accuracy to being male (vs. female), being old (vs. young), and to seeing an old (vs. young) target face. Being old and the target face being masked jointly predicted the lowest probability of accurate emotion recognition. Older adults were expected to misclassify more than 50% of the emotional expressions for masked targets. This finding lends support to the idea that aging-related changes in brain structure and neurotransmitter balance are particularly consequential when circumstances are

challenging [e.g., 28]. Aging-related changes may deprive older adults of resources necessary to adapt to difficulties posed by the experimental task. However, contrasting this explanation focused on neurological change, motivational differences between younger and older adults may also play a role. With increasing age, individuals' time horizon shrinks, leading to motivational changes [i.e., increased importance of positive experiences; 61,62]. Because older adults are relatively more motivated than younger adults to enhance their emotional well-being [63], the uncertainty regarding the target's emotional expression due to being masked may afford them to make goal-congruent emotion attributions [e.g., 64]. In other words, the mask-induced ambiguity of the expressed emotion may allow older adults to see what they want to see, resulting in lower overall accuracy. Future research may attempt to disentangle the relative contribution of neurological and motivational factors.

## Face masks and social judgments

In our initially planned analyses, we found that wearing a face mask (vs. not) by itself did not affect social judgments. This finding is inconsistent with the results of a recent study by Cartaud and colleagues [65] who found a positive effect of face masks (vs. no face masks) on perceived trustworthiness. Nonetheless, when controlling for the expressed emotion's valence and mask-related associations, face masks (vs. no face masks) predicted lower perceptions of closeness. Moreover, face masks interacted with the valence of the expressed emotion in shaping all three social judgements. For unmasked faces, negative (vs. non-negative) emotional expressions led to lower ratings of trustworthiness, likability, and closeness. This is in line with the emotion-as-social-information model [6], which holds that observers use others' facial emotional expressions to make inferences about them. As negative emotions generally serve social disengagement [47, but see also 48], observers may interpret negative emotional expressions accordingly (as reflected in lower ratings of trustworthiness, likability, and closeness; this study). Strikingly, face masks buffered these negative effects of expressing negative (vs. non-negative) emotions on and across the three social judgments. While participants rated targets to be less trustworthy, less likable, and less close when expressing negative (vs. non-negative) emotions, they did so to a smaller extent for masked (vs. unmasked) faces.

As face masks increase the ambiguity of the expressed emotion, they may leave room for a positivity bias to shape judgments [66]. Hence, the uncertainty related to targets' wearing face masks appears not so much to impact social judgments directly but rather indirectly. Whether the moderating influence of face masks on the relationship between negative (vs. non-negative) emotional expressions and social judgments can be considered adaptive or maladaptive is unclear. On the one hand, face masks' buffering effect corresponds to an impairment of people's ability to appropriately respond to social threats. While negative (vs. non-negative) emotional expressions facilitate social disengagement [47, but see also 48], our results indicate that face masks may undermine this function. Therefore, improved social judgments for masked (vs. unmasked) targets may be a maladaptive consequence of face masks. On the other hand, face masks' buffering effect may be adaptive. If people deem masked persons expressing distress or disapproval to be more trustworthy, likable, and close, they may be more likely to offer support, a putatively adaptive consequence [67].

In addition to valence's and face masks' effect, mask-as-threat associations influenced perceived closeness. For masked but not for unmasked faces, associating face masks more strongly with threats predicted higher perceived closeness. Because masked others apparently acknowledge the virus' danger and take measures against it, people with strong mask-as-threat associations may consider masked targets as similar, leading to higher perceptions of closeness in turn [68].

In addition to being predicted to perceive a masked (vs. unmasked) other as closer, individuals who strongly associated face masks with threats tended to also associate them more strongly with opportunities. As individuals seek out more information about the virus, its consequences, and protective measures, they do not only learn more about the dangers of COVID-19 but also about the merit of face masks. This interpretation is line with the finding (see the online supplemental material) that mask-as-threat and mask-as-opportunity associations tend to be stronger for individuals who are more preoccupied with COVID-19 (mask-as-threat: $r = .52$, $p < .001$; mask-as-opportunity: $r = .32$, $p < .001$).

## Practical implications

Face masks curtail emotion-recognition accuracy and perceptions of closeness. These side effects may be particularly worrisome in settings where accurate emotion recognition and establishing relationships is pivotal (e.g., health and financial sectors). Emotions regulate interpersonal interactions by instigating inferential processes, affording interaction partners with a set of appropriate behaviors [6,69]. For example, the appropriateness of making another joke can be deduced from the emotions which others expressed in response to the first joke (e.g., anger or joy). Hence, by reducing the likelihood of accurate emotion recognition, face masks may undermine the success of our social interactions. Moreover, mimicking others' emotional expressions serves affiliation goals and increases liking [13,14]. Given that face masks reduce emotion-recognition accuracy, observer may mimic an emotion which was not expressed by a masked target. Mimicking the wrong emotion may give rise to perceptions of dissimilarity and non-affiliative intents, ultimately frustrating affiliation goals [70,71]. Moreover, emotion recognition is a subcomponent of empathy and predicts prosocial behavior via empathy [72,73]. Therefore, to the extent that emotion recognition is reduced, prosocial behavior may be less likely.

Face masks may further undermine adherence to social distancing. Because face masks limit facial cues, people may compensate by approaching each other. Reductions in facial cues were shown to result in less physical distance between interaction partners [65,74]. Hence, governmental entities may want to actively encourage alternative ways of compensating for the loss of facial cues due to wearing a face mask like amplifying facial emotion expressions (especially in the eye region) or using body language and clear emotion-focused language.

Emotion recognition is also crucial for deceit detection [75]. Therefore, face masks may increase fraud susceptibility. Older adults may be especially vulnerable given their low emotion-recognition accuracy for masked faces (this study). Hence, older adults may particularly benefit from interventions focused on improving emotional decoding or on information about common scams [76].

## Generalizability, limitations, and future directions

We demonstrated that face masks influence emotional inferences and social judgments. Nevertheless, some limitations apply. As face masks were mandatory during the period of data collection, precautions are warranted when generalizing the results to regions without mandatory face masks in public, or regions where face masks are heavily politicized. At least for perceived closeness, we demonstrated that mask-related associations matter. Precautions are also warranted when generalizing the results to other cultural or demographic groups. Regarding the former groups, cultural differences in emotional accents [akin to accented speech; 60,77] or display rules [78] result in variation in emotion expression. Importantly, display rules prohibiting (vs. allowing) the forthright expression of emotions [e.g., 78] or unfamiliarity (vs. familiarity) with a specific emotional accent (e.g., tourists or immigrants) may place an additional

burden on observers by restricting the already limited number of facial cues in masked faces. In such contexts, the negative effect of face masks on emotion-recognition accuracy may be more pronounced. Regarding the latter groups, children exhibit a preference for the eyes, a region not covered by face masks, when processing faces [79]. This preference may translate into better emotion-recognition accuracy relative to adults. Future research may examine these possibilities.

While the use of static portrait photographs as stimuli is common, facial expressions are naturally dynamic. Faces' dynamic features convey important information about emotional states [80]. Therefore, future research may extend our findings by exploring face masks' effect on emotional inferences and social judgments for dynamic facial expressions.

Most of the target expressions in this study are negatively valenced (sadness, anger, fear, and disgust vs. happiness and neutral). Yet, positively-valenced emotions are not only actively sought after in daily life [e.g., 81] but are also multifaceted [e.g., 18,82]. Hence, future research may extend our findings by considering a variety of positively-valenced emotional expression and their interactions with face masks.

## Conclusion

While face masks effectively curb the spread of COVID-19 [3], they have collateral consequences for emotional inferences and social judgments. Face masks impair people's ability to accurately classify emotional expressions which tends to be exacerbated for older adults, reduce perceived closeness, and increase perceptions of closeness, trustworthiness, and likability for targets expressing negative emotions. Together, our results highlight the relevance of psychological factors in the context of the coronavirus and inform policymaking by illustrating face masks' (side) effects on social functioning.

## Author Contributions

**Conceptualization:** Felix Grundmann, Kai Epstude, Susanne Scheibe.

**Data curation:** Felix Grundmann.

**Formal analysis:** Felix Grundmann.

**Investigation:** Felix Grundmann.

**Methodology:** Felix Grundmann, Kai Epstude, Susanne Scheibe.

**Project administration:** Felix Grundmann, Kai Epstude, Susanne Scheibe.

**Supervision:** Kai Epstude, Susanne Scheibe.

**Visualization:** Felix Grundmann.

**Writing – original draft:** Felix Grundmann.

**Writing – review & editing:** Felix Grundmann, Kai Epstude, Susanne Scheibe.

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
