## [Decision Letter · Decision Letter 0]

9 Feb 2021

PONE-D-21-00370

Face masks reduce emotion-recognition accuracy and perceived closeness

PLOS ONE

Dear Dr. Grundman:

Thank you for submitting your manuscript to PLOS ONE. After careful consideration, we feel that it has merit but does not fully meet PLOS ONE’s publication criteria as it currently stands. Therefore, we invite you to submit a revised version of the manuscript that addresses the points raised during the review process.

I have now reportd of two experts in the field (you can find them bvelow). Both of them find interest in your work, but also express several concerns that should be carefully addressed before the MS will meet publoication.

I also had a close look at your work. My suggestion would be to discuss possible cultural limitations (see, e.g., previous work of my own section Pavlova et al., 2018). PLease adress all concerns point by poiui in your rebuttal letter to me.

We look forward to receiving your revised manuscript.

Kind regards,

Marina A. Pavlova, PhD

Academic Editor

PLOS ONE

Journal Requirements:

2. We note that Figure 1 in your submission contain copyrighted images. All PLOS content is published under the Creative Commons Attribution License (CC BY 4.0), which means that the manuscript, images, and Supporting Information files will be freely available online, and any third party is permitted to access, download, copy, distribute, and use these materials in any way, even commercially, with proper attribution. For more information, see our copyright guidelines: http://journals.plos.org/plosone/s/licenses-and-copyright.

2.1.    You may seek permission from the original copyright holder of Figure 1 to publish the content specifically under the CC BY 4.0 license.

2.2.    If you are unable to obtain permission from the original copyright holder to publish these figures under the CC BY 4.0 license or if the copyright holder’s requirements are incompatible with the CC BY 4.0 license, please either i) remove the figure or ii) supply a replacement figure that complies with the CC BY 4.0 license. Please check copyright information on all replacement figures and update the figure caption with source information. If applicable, please specify in the figure caption text when a figure is similar but not identical to the original image and is therefore for illustrative purposes only.

Reviewers' comments:

Reviewer's Responses to Questions

**Comments to the Author**

1. Is the manuscript technically sound, and do the data support the conclusions?

Reviewer #1: Yes

Reviewer #2: Yes

2. Has the statistical analysis been performed appropriately and rigorously? 

Reviewer #1: Yes

Reviewer #2: Yes

3. Have the authors made all data underlying the findings in their manuscript fully available?

Reviewer #1: Yes

Reviewer #2: Yes

4. Is the manuscript presented in an intelligible fashion and written in standard English?

Reviewer #1: Yes

Reviewer #2: Yes

5. Review Comments to the Author

Reviewer #1: Authors tested the hypothesis that face masks affect emotion recognition and social judgments by performing emotion recognition tasks presenting masked vs not masked faces expressing primary emotions, exploring also the role of additional variables such as subject age, age of the human observed, the gender of the subject, etc...

The topic of the study is very relevant since it focuses on the consequences of COVID-19 related PPE on social cognition. I found the study technically sound. I report here some concerns to be addressed:

1- INTRODUCTION: I suggest reporting hypotheses at the end of the introduction section.

2- METHOD: I’m not fully convinced about the choice of authors to add three distractors in the response alternatives. My concern is related to the fact that no items represent the “amused/proud/surprised” condition. Also, amused and proud are high order emotion, and they amply differed from the other alternatives. Finally, in a certain sense, amused could be interpreted as a level of happiness, and I’m not convinced that the error of a choice “amused” instead of “happy” and the choice “neutral” instead of “happy” reflect the same type error.

3- RESULTS: In results line 242 to 245 authors report some trends of difference and significant difference, I suggest adding p-values.

At line 295 I suggest reporting the ICC (contingency).

I found unexpected the significant correlation between mask-as-opportunity and mask-as-threat. How author discuss these results?

Reviewer #2: In the context of the present pandemic context, the present paper questions the effect of face masks on emotion-recognition accuracy and social judgments (perceived trustworthiness, likability, and closeness). The study (N=191, Germany) revealed that face masks diminish people’s ability to accurately categorize emotion expression and make target persons appear less close. Exploratory analyses further revealed that face masks buffered the effect of negative emotion expressions on perceptions of trustworthiness, likability, and closeness. The results revealed higher perceptions of closeness for masked but not for unmasked faces, which could represent a valuable information for policymakers.

This study examined face masks’ effect on emotional and social inferences. The authors investigated whether the reduction of facial cues due to wearing a face mask undermines emotion-recognition accuracy and perceptions of social important traits such as trustworthiness, likability, and closeness. They specifically explored how the valence of the emotional expression and mask-related social judgments interact. The study is appealing, well conducted and provide important results. The data are appropriately and thoroughly analyzed. The data show that emotion-recognition accuracy declines with masked faces. Furthermore, lower accuracy was found with male (vs. female), old (vs. young) participants, as well as when seeing an old (vs. young) face. The worst case was when older participants were facing masked faces (below 50% of success). However, wearing a face mask did not affect social judgments (excepted closeness). An interesting result is that face mask lowers the effect of negative emotions, with the risk of undermining their social relevance in particular their role in prosocial behaviour. The paper is well written. My only concern is that the study involved more negative (4) than positive (1) emotions, which represent a limit of the study. Hereafter are detailed my main comments.

Introduction

- The concern of trustworthiness and perceived health of faces wearing face mask has already being the subject of investigation and publication, please refer to Cartaud A, Quesque F, Coello Y (2020) Wearing a face mask against Covid-19 results in a reduction of social distancing. PLoS ONE 15(12): e0243023. https://doi.org/10.1371/journal.pone.0243023.

- L. 111. When stating “Indeed, different studies have linked familiarity, ambiguity, or uncertainty to lower perceptions of trustworthiness, likability, and closeness”, references are required.

- Were participants really rewarded 1€ ?

- The participants were evaluated on mask-as-threat and mask-as-opportunity associations, preoccupation with COVID-19 and exposure to face masks. Could you provide information about how this was done (even if it is not relevant to the study)? Where the supplementary information can be found is not indicated and not available on the PlosOne website. (This also applies to emotion codes and reliability estimations)

Method

- Concerning the stimuli in each group, each emotion was presented once in each condition (age (3)*gender(2)*emotion(6)), which is quite low. Is there any reason for selecting this restricted set of stimuli (36 is not much) ?

- The use of emotion distractors is a good idea. However, I was wondering whether “happy” and “amused” are not too close from each other. Were they any sign of confounds in the data? Why not having chosen only abstract category such as proud for instance, for which no facial expression is objectively associated? (This also applies to surprise, which is an emotion, unlike proud).

- Figure 1. Response items are not visible in frame 2 and 3 is not visible. Could the Figure be improved?

- The 3 statements used to evaluate « mask-as-threat » and « mask-as opportunity » associations should be provided in the text.

Data analysis

- It is not clear why “adding interactions between condition and age as well as between condition and age-of-target improved the fit of the model considerably, χ2262 (4) = 8.42, p = .074” since the statistic is not significant.

- It is surprising that the level of emotion recognition remains high with the mask (48.9%), despite the lack of crucial emotional cues. Could this outcome be the consequence that the number of negative emotions (4) was high compared to the number of positive emotions (1), which are known to be detected mostly on the upper part of the face.

Discussion

- L. 349. The quoted reference (53) does not seem to be the correct one.

- it is indicated the “overall accuracy for unmasked as well as masked targets was considerably lower than previous study”. This could be due to the fact that you used also elderly faces in the present study. Can you comment?

- The observation that wearing a face mask did not affect social judgments should be discussed in regards to the previous study published by Cartaud, Quesque, & Coello (2020).

- The authors suggest that “Face masks may further undermine adherence to social distancing. Because face masks limit facial cues, people may compensate by approaching each other". This has been indeed shown in previous study and should be indicated/discussed (Cartaud, Quesque, & Coello, 2020).

6. PLOS authors have the option to publish the peer review history of their article (what does this mean?). If published, this will include your full peer review and any attached files.

Reviewer #1: No

Reviewer #2: No

---

## [Author Response · Author response to Decision Letter 0]

1 Mar 2021

RESPONSES TO THE EDITOR

• Comment 1: My suggestion would be to discuss possible cultural limitations (see, e.g., previous work of my own section Pavlova et al., 2018).

o Authors’ response: We appreciate the editor’s suggestion and agree that a discussion of possible cultural limitations would benefit the manuscript. In the previous version of the manuscript, we discussed our findings in relation to demographic differences (i.e., age; LL 478-480). We implemented the editor’s suggestion by also discussing the role of cultural differences. Specifically, we shortly outline how cultural differences in emotional accents and display rules may influence the observed effect of face masks (vs. no face masks) on emotion-recognition accuracy (LL 472-478).

RESPONSES TO REVIEWER 1

• Comment 1: Authors tested the hypothesis that face masks affect emotion recognition and social judgments by performing emotion recognition tasks presenting masked vs not masked faces expressing primary emotions, exploring also the role of additional variables such as subject age, age of the human observed, the gender of the subject, etc...

The topic of the study is very relevant since it focuses on the consequences of COVID-19 related PPE on social cognition. I found the study technically sound. I report here some concerns to be addressed:

I suggest reporting hypotheses at the end of the introduction section

o Authors’ response: We thank the reviewer for the suggestion. Because we report confirmatory as well as exploratory analyses, ending the introduction section with remarks on exploratory steps, we believe that reporting all hypotheses at the end of the introduction section may be confusing for the reader. To remind themselves of the rationale underlying each hypothesis, readers would have to go back to earlier paragraphs. Moreover, the hypotheses pertain to different outcomes (emotion-recognition accuracy and social judgements). Hence, reporting them together may not be ideal. To accommodate the reviewer’s suggestion, we report the two sets of hypotheses (grouped by outcome) at the end of two respective sections (see LL 94-102 for emotion-recognition accuracy and LL 127-130 for the social judgements).

• Comment 2: My concern is related to the fact that no items represent the “amused/proud/surprised” condition. Also, amused and proud are high order emotion, and they amply differed from the other alternatives. Finally, in a certain sense, amused could be interpreted as a level of happiness, and I’m not convinced that the error of a choice “amused” instead of “happy” and the choice “neutral” instead of “happy” reflect the same type error.

o Authors’ response: The reviewer made several important points. To address the remark that amusement reflects a level of happiness, we performed additional analyses in which we treated responses in which participants classified a happy face as being amused as correct responses. We report the results of these analyses in the online supplemental material (see LL 244-246). Even when considering judgements of amusement for happy target faces as correct, the results were largely unchanged. We still observe a strong effect of condition as well as effects of gender and age of target (young vs. old) for the main-effects model and for the model including interaction effects between condition and age and between condition and age of target. The only difference emerged for the effect of age (young vs. old) on emotion-recognition accuracy which only reached statistical significance in the main-effects model. Consistent with the preregistration, we focus in the manuscript on the analyses in which we only treat correct judgment-emotion pairings as correct classifications. We do so for two reasons. First, amusement can be decoded based on facial cues alone (see newly added references 18 & 60). Second, according to our theoretical framework, expressed emotions convey information important for social functioning. Although different expressions may share the same valence or are similarly expressed, their meanings and related implications for (interpersonal) functioning may differ. Hence, correct classifications (defined as correct judgement-emotion pairings) are crucial.

o We agree with the reviewer that not having targets expressing amusement, pride, or surprise may be a point of concern. Participants might have learned to classify every positively-valenced target expression as happiness because amusement and pride (and surprise) were never expressed. We now discuss this possibility in LL 381-384. Nevertheless, we believe that our results constitute a better representation of emotion-recognition accuracy in everyday life compared to a similar study using the same stimuli but no distractor items (see reference 20). In this study, the risk of participants learning to classify each positively-valenced target expression as happiness was relatively greater, posing a greater threat to external validity in turn (see also LL 376-381). 

o We also agree with the reviewer that pride and amusement are higher-order emotions. Given the purpose of the present study, it was critical that all emotions are expressed in distinct ways in the face. This is the case for pride and amusement (see references 18 & 60). Moreover, we included distractor items to address potential ceiling effects for positively-valenced emotional expression. For example, in a recent study which included the same stimuli but no distractor items, emotion-recognition accuracy for happy target expressions approached 100% (see reference 20). Similarly, including the distractor items in the current study reduced the chance that participants simply learned to classify a positively-valenced face as expressing happiness based on the lack of alternative classification options.

• Comment 3: In results line 242 to 245 authors report some trends of difference and significant difference, I suggest adding p-values.

o Authors’ response: We appreciate the reviewer’s suggestion and incorporated it by adding Figure 2 (LL 263-264, 276-280). In Figure 2, we illustrate the observed differences in emotion-recognition accuracy (across participants and trials) based on condition, age, gender, and age of target. P-values are also included in the figure. We computed the p-values by fitting simple multilevel-logistic regression models with emotion-recognition accuracy as the outcome and the factor of interest as the sole predictor. For factors with three levels, we repeated the analysis using the level initially coded as 010 as the reference group.

• Comment 4: At line 295 I suggest reporting the ICC (contingency).

o Authors’ response: We thank the reviewer for the suggestion. Assuming ICC refers to the intra-class correlation, we added the ICC in LL 280 & 303. We also moved the explanation of what the ICC means to LL 230-232 and added reference 57.

• Comment 5: I found unexpected the significant correlation between mask-as-opportunity and mask-as-threat. How author discuss these results?

o Authors’ response: Consistent with the reviewer’s comment, we included a discussion of the positive correlation between mask-as-threat and mask-as-opportunity associations (see LL 434-440). In short, individuals who actively search for information about COVID-19 and frequently think and talk about it tend to report stronger mask-related associations. Hence, the association may indicate that individuals who better inform themselves about COVID-19, learn more about both its risks and the opportunities afforded by wearing face masks. 

RESPONSES TO REVIEWER 2

• Comment 1.1: In the context of the present pandemic context, the present paper questions the effect of face masks on emotion-recognition accuracy and social judgments (perceived trustworthiness, likability, and closeness). The study (N=191, Germany) revealed that face masks diminish people’s ability to accurately categorize emotion expression and make target persons appear less close. Exploratory analyses further revealed that face masks buffered the effect of negative emotion expressions on perceptions of trustworthiness, likability, and closeness. The results revealed higher perceptions of closeness for masked but not for unmasked faces, which could represent a valuable information for policymakers.

This study examined face masks’ effect on emotional and social inferences. The authors investigated whether the reduction of facial cues due to wearing a face mask undermines emotion-recognition accuracy and perceptions of social important traits such as trustworthiness, likability, and closeness. They specifically explored how the valence of the emotional expression and mask-related social judgments interact. The study is appealing, well conducted and provide important results. The data are appropriately and thoroughly analyzed. The data show that emotion-recognition accuracy declines with masked faces. Furthermore, lower accuracy was found with male (vs. female), old (vs. young) participants, as well as when seeing an old (vs. young) face. The worst case was when older participants were facing masked faces (below 50% of success). However, wearing a face mask did not affect social judgments (excepted closeness). An interesting result is that face mask lowers the effect of negative emotions, with the risk of undermining their social relevance in particular their role in prosocial behaviour. The paper is well written. 

My only concern is that the study involved more negative (4) than positive (1) emotions, which represent a limit of the study. Hereafter are detailed my main comments.

o Authors’ response: We thank the reviewer for sharing their concern. To address it, we discuss the imbalance between positively- and negatively-valenced emotional expressions in the present study as a limitation (see LL 486-490)

• Comment 2: The concern of trustworthiness and perceived health of faces wearing face mask has already being the subject of investigation and publication, please refer to Cartaud A, Quesque F, Coello Y (2020) Wearing a face mask against Covid-19 results in a reduction of social distancing. PLoS ONE 15(12): e0243023. https://doi.org/10.1371/journal.pone.0243023.

o Authors’ response: We thank the reviewer for bringing this recent publication to our attention. Given its relevance, we cite it in LL 403-405 & 456-457.

• Comment 3: L. 111. When stating “Indeed, different studies have linked familiarity, ambiguity, or uncertainty to lower perceptions of trustworthiness, likability, and closeness”, references are required.

o Authors’ response: Consistent with the reviewer’s comment, we added references 34-36 (see LL 110-112). 

• Comment 4: Were participants really rewarded 1€ ?

o Authors’ response: We commissioned the panel company Respondi to collect the data. They decided on the reimbursement of its panel members (1€) based on the estimated duration of the survey (20 minutes). 

• Comment 5: The participants were evaluated on mask-as-threat and mask-as-opportunity associations, preoccupation with COVID-19 and exposure to face masks. Could you provide information about how this was done (even if it is not relevant to the study)? Where the supplementary information can be found is not indicated and not available on the PlosOne website. (This also applies to emotion codes and reliability estimations)

o Authors’ response: We agree with the reviewer that the reader may be interested to know how the constructs were assessed. Therefore, we now provide the requested information in LL 175-176. We also added a link redirecting the reader to the storage location of the online supplemental material when mentioning it for the first time (see LL 177-179).

• Comment 6: Concerning the stimuli in each group, each emotion was presented once in each condition (age (3)*gender(2)*emotion(6)), which is quite low. Is there any reason for selecting this restricted set of stimuli (36 is not much)?

o Authors’ response: We agree that the number of stimuli may appear limited relative to studies such as the one conducted by Cartaud and colleagues (2020; mentioned in Comment 2). Yet, our final level-1 sample size was nevertheless large enough to detect meaningful effects (N = 6874-6876, depending on missing values). Please also note that when we designed the study, we 1) sought to not overburden participants (assuming trial completion time is 20 seconds, completing all trials still took 12 minutes) and 2) had to consider financial constraints (participants’ reimbursement was based on completion time).

• Comment 7: - The use of emotion distractors is a good idea. However, I was wondering whether “happy” and “amused” are not too close from each other. Were they any sign of confounds in the data? Why not having chosen only abstract category such as proud for instance, for which no facial expression is objectively associated? (This also applies to surprise, which is an emotion, unlike proud).

o Authors’ response: We opted for amusement and pride as distractor items because they are associated with distinct patterns of facial expression (see references 18 & 60). Similarly, individuals associate distinct facial cues with the expression of surprise (e.g., Elfenbein & Ambady, 2002; Russell, 1994), with the additional benefit of being relevant for positively- as well as negatively-valenced expressions (e.g., Noordewier & van Dijk, 2019). 

- Elfenbein, H. A., & Ambady, N. (2002). On the universality and cultural specificity of emotion recognition: A meta-analysis. Psychological Bulletin, 128(2), 203–235. https://doi.org/10.1037/0033-2909.128.2.203.

- Noordewier, M. K., & van Dijk, E. (2019). Surprise: Unfolding of facial expressions, Cognition and Emotion, 33(5), 915–930. https://doi.org/10.1080/02699931.2018.1517730

- Russell, J. A. (1994). Is there universal recognition of emotion from facial expression? A review of the cross-cultural studies. Psychological Bulletin, 115(1), 102–141. https://doi.org/10.1037/0033-2909.115.1.102

o We appreciate the reviewer’s question about possible confounds. Alluding to the other comments of the reviewer, the choice of distractor items might have influenced the results. However, we are unsure how and why this should have been the case. Hence, we do not believe that any confounds threaten the internal validity of the results. Nonetheless, we acknowledge that the impact of distractor-item choice on emotion-recognition accuracy constitutes an empirical question yet to be answered. Moreover, we highlight a possible limitation related to the fact that choosing a distractor item was never correct (see LL 381-384).

o We further thank the reviewer for the thoughtful remark concerning the resemblance of expressions of happiness and amusement. Reviewer 1 voiced a similar concern, which we addressed above (see Comment 2). In short, we performed additional analyses in which we treated judgements of amusement for happy target faces as correct classifications. We report the results in the online supplemental material which are largely unchanged. Generally, we decided to adhere to our preregistered analyses plan because 1) amusement is characterized by distinct facial cues (see references 18 & 60) and 2) our theoretical framework suggests that correctly decoding emotional expressions of others is crucial for social functioning (even if expressions are similar).

• Comment 8: Figure 1. Response items are not visible in frame 2 and 3 is not visible. Could the Figure be improved?

o Authors’ response: Consistent with the reviewer’s comment, we improved Figure 1 by increasing the visibility of the response items. We further noted that the item wording is not accurate (see L 191), as the study was conducted in German.

• Comment 9: The 3 statements used to evaluate « mask-as-threat » and « mask-as opportunity » associations should be provided in the text.

o Authors’ response: As suggested by the reviewer, we provide all statements used to measure mask-as-threat and mask-as-opportunity associations in the text (see LL 211-217) 

• Comment 10: It is not clear why “adding interactions between condition and age as well as between condition and age-of-target improved the fit of the model considerably, χ2 (4) = 8.42, p = .074” since the statistic is not significant.

o Authors’ response: We agree with the reviewer that our formulation may be somewhat misleading. To improve the sentence, we replaced ‘considerably’ with ‘marginally’ (see LL 287-290). 

• Comment 11: It is surprising that the level of emotion recognition remains high with the mask (48.9%), despite the lack of crucial emotional cues. Could this outcome be the consequence that the number of negative emotions (4) was high compared to the number of positive emotions (1), which are known to be detected mostly on the upper part of the face.

o Authors’ response: We appreciate the reviewer’s comment on the overall emotion-recognition accuracy in the mask-condition. It could indeed be possible that the recognition rate is partly the result of our choice of target expressions. Another explanation, also focused on the nature of the expressions, may be that the decoding of basic emotions (fear, anger, disgust, happiness, sadness) is an automatic and evolutionary relevant process. Hence, the decoding of emotional expressions is expected to still be possible even if facial cues are limited. At this point, we also feel the need to caution readers against interpretating the overall emotion-recognition accuracy in the mask condition as either high or low. We do so because one could also consider the recognition rate as low. In a recent study which used the same stimuli but no distractor items (see reference 20), emotion-recognition accuracy was considerably higher in the mask condition (~71.7). This difference is surprising, as we collected our data at least 6 weeks later (data files of Carbon’s study were uploaded to the open science framework on May 23, 2020, while we collected our data between June 25, 2020, and July 8, 2020). As face masks were (and still are) mandatory in public at the time, participants in the present study were more familiar with masked others and had more time to practice recognizing emotions in masked faces. This would lead one to intuitively expect that participants in the study which was conducted later (this study) recognize emotions in masked faces with greater accuracy. Yet, emotion-recognition accuracy is lower in the present study.

• Comment 12: L. 349. The quoted reference (53) does not seem to be the correct one.

o Authors’ response: The reviewer correctly pointed out that the quoted reference was incorrect. We added appropriate references (see references 59-60).

• Comment 13: L. 349. it is indicated the “overall accuracy for unmasked as well as masked targets was considerably lower than previous study”. This could be due to the fact that you used also elderly faces in the present study. Can you comment?

o Authors’ response: We thank the reviewer for the comment. In the manuscript, we argue that the disparity in emotion-recognition accuracy is due to our inclusion of distractor items (see LL 376-381). Given that the cited study (reference 20) used the same stimuli (including elderly faces) but no distractor items, we believe that it is likely that the inclusion of distractor items at least in part led to the difference in recognition rates.

• Comment 14: L. The observation that wearing a face mask did not affect social judgments should be discussed in regards to the previous study published by Cartaud, Quesque, & Coello (2020).

o Authors’ response: We agree that the results reported by Cartaud and colleagues (2020) should be highlighted when we discuss our findings. We did so in LL 403-405. We refrained from a more elaborate discussion of the relationship between our and Cartaud and colleagues’ (2020) findings, as this would have resulted in a shift of the manuscript’s focus. As it is now, the primary focus of the manuscript lies on the buffering effect of face masks in the context of negative emotional expressions’ effect on the social judgements.

• Comment 15: L. The authors suggest that “Face masks may further undermine adherence to social distancing. Because face masks limit facial cues, people may compensate by approaching each other". This has been indeed shown in previous study and should be indicated/discussed (Cartaud, Quesque, & Coello, 2020).

o Authors’ response: We agree that the reference mentioned by the reviewer is relevant and thus integrated it (see LL 456-457).

---

## [Decision Letter · Decision Letter 1]

25 Mar 2021

Face masks reduce emotion-recognition accuracy and perceived closeness

PONE-D-21-00370R1

Dear Dr. Grundman,

We’re pleased to inform you that your manuscript has been judged scientifically suitable for publication and will be formally accepted for publication once it meets all outstanding technical requirements.

Kind regards,

Marina A. Pavlova, PhD

Academic Editor

PLOS ONE

Additional Editor Comments (optional):

Reviewers' comments:

Reviewer's Responses to Questions

**Comments to the Author**

1. If the authors have adequately addressed your comments raised in a previous round of review and you feel that this manuscript is now acceptable for publication, you may indicate that here to bypass the “Comments to the Author” section, enter your conflict of interest statement in the “Confidential to Editor” section, and submit your "Accept" recommendation.

Reviewer #1: All comments have been addressed

Reviewer #2: All comments have been addressed

2. Is the manuscript technically sound, and do the data support the conclusions?

Reviewer #1: Yes

Reviewer #2: Yes

3. Has the statistical analysis been performed appropriately and rigorously? 

Reviewer #1: Yes

Reviewer #2: Yes

4. Have the authors made all data underlying the findings in their manuscript fully available?

Reviewer #1: Yes

Reviewer #2: Yes

5. Is the manuscript presented in an intelligible fashion and written in standard English?

Reviewer #1: Yes

Reviewer #2: Yes

6. Review Comments to the Author

Reviewer #1: The authors addressed all my concerns and I believe that the manuscript is now suitable to be published.

Reviewer #2: The authors addressed all the comments and I am satisfied with the answers they provided and the manuscript improvement.

7. PLOS authors have the option to publish the peer review history of their article (what does this mean?). If published, this will include your full peer review and any attached files.

Reviewer #1: No

Reviewer #2: No

---

## [Editor Report · Acceptance letter]

13 Apr 2021

PONE-D-21-00370R1 

Face masks reduce emotion-recognition accuracy and perceived closeness 

Dear Dr. Grundmann:

I'm pleased to inform you that your manuscript has been deemed suitable for publication in PLOS ONE. Congratulations! Your manuscript is now with our production department. 

Kind regards, 

on behalf of

Prof. Marina A. Pavlova 

Academic Editor

PLOS ONE